# Developing New Method in Measuring City Economic Resilience by Imposing Disturbances Factors and Unwanted Condition

**Titi Purwandari** [1,*], **Sukono** [2], **Yuyun Hidayat** [3] **and Wan Muhamad Amir W. Ahmad** [4]

1   Doctoral Program of Mathematics, Faculty of Mathematics and Natural Sciences, Universitas Padjadjaran, Jatinangor 45363, West Java, Indonesia
2   Department of Mathematics, Faculty of Mathematics and Natural Sciences, Universitas Padjadjaran, Jatinangor 45363, West Java, Indonesia
3   Department of Statistics, Faculty of Mathematics and Natural Sciences, Universitas Padjadjaran, Jatinangor 45363, West Java, Indonesia
4   School of Dental Sciences, Health Campus, Universiti Sains Malaysia, Kubang Kerian 16150, Kelantan, Malaysia
*   Correspondence: t.purwandari@unpad.ac.id

**Abstract:** Recent research uses an index to measure economic resilience, but the index is inadequate because it is impossible to determine which disturbance factors have the greatest impact on the economic resilience of cities. This study aims to develop a new methodology to measure the economic resilience of a city by simultaneously examining unwanted conditions and disturbance factors. The ratio of regional original income to the number of poor people is known as Z and is identified as a measure of economic resilience in Indonesia. Resilience is measured by Z's position in relation to the unwanted area following a specific level of disturbance. If Z is in the unwanted condition, the city's per capita income will decrease, and the city will be considered economically not resilient. The results of the analysis show that six levels of economic resilience have been successfully distinguished based on research on 514 cities in Indonesia involving nine indicators of disturbance and one variable of economic resilience during the five-year observation period, 2015–2019. Only 3.11 percent of cities have economic resilience level 1, while 69.18 percent have level 0. Economically resilient cities consist of 4.24 percent of cities at level 2, as much as 3.39 percent at level 3, as much as 3.39 percent at level 4, and as much as 16.69 percent at level 5. The novelty of this research is to provide a new methodology for measuring the economic resilience of cities by integrating unwanted conditions as necessary conditions and disturbance factors as sufficient conditions. The measurement of a city's economic resilience is critical to help the city government assess the security of the city so the government can take preventive actions to avoid the cities falling into unwanted conditions.

**Keywords:** economic resilience; unwanted conditions; disturbance factors

## 1. Introduction

Resilience is an "ecological" term that focuses on a region's ability to absorb a significant amount of disturbance before deviating from its initial state. In the Government of Indonesia Work Plan 2021, strengthening economic resilience for quality and fair growth is highlighted as a national priority [1].

Research to throw light on some of the underlying aspects of regional resilience and provide an overview of the notion, as well as an analysis of the research studies on constructing the territorial composite indices (CIs), was also carried out by [2]. The study reported that CI construction suffers from many methodological difficulties, with the result that they can be misleading and easily manipulated. The analyses of economic resilience and its main determinant factors of the regions of seven Eastern European countries [3] conclude that the determining factors of resilience for the studied regions concern the size

of the manufacturing sector, the services and public administration, and entrepreneurship and human capital represented by tertiary education; agriculture and urban population have no significant influence on regional resilience.

Currently, Indonesia and many researchers in other countries have chosen to use the index to measure the status of its economic resilience. The results indicate that Indonesia's economic resiliency is in good condition [4]. The resilience measurement using the index is not carried out only by Indonesia. The number of indexes in existence around the world is growing every year [5–7]. The following descriptions show that the measurement of economic resilience using index numbers is still common practice internationally. The economic resilience index uses microeconomic, mesoeconomic, and macroeconomic variables, and the results of the study obtained the framework for determining an economic resilience index developed by [8]. Research on regional economic resilience in the face of natural disasters by establishing a resistance and recovery index is also carried out [9]. This study discusses a country which was hit by a major earthquake, while the factors studied were regional demographics, economic aspects, and regional housing values, using the resistance index and sensitivity index.

This study contradicts previous studies that used an index-based method to determine economic resilience. This is because an index number is unable to answer the question, "resistant to what?" The economic resilience index simply states that a higher index value indicates that a city is becoming more resilient over time. Our earlier research prepared a comprehensive explanation of the index's deficiencies in measuring the city's economic resilience [7,10,11]. The substantial meaning of resilience raises a relevant and fundamental research question, namely "resilience to what?" To disturbances, of course. The strategic research question is the following: "how strong must a city be to withstand the effect of an economic shock, and at what intensity?" The next question is: "what are the criteria for a city's economic resilience in the context of the disturbance?" That question cannot be solved by the index, but it can be solved by a model that shows the relationship between the disturbance factor and the unwanted condition. The unwanted condition is not explicitly defined in any publication. Many researchers' approach to measuring economic resilience by merely defining an index inspires writers to propose a different technique. The economic resilience of a city requires the existence of unwanted conditions and a level of disturbance factors; this is achieved by integrating previous studies [7,10,11].

The novelty of this research is to provide a new methodology for measuring the economic resilience of cities by integrating unwanted conditions as necessary conditions, and disturbance factors as sufficient conditions. The ability to measure a city's economic resilience is critical for recognizing factors that could cause the city to collapse.

The findings of this study are aimed toward improving a country's economic development. Policymakers can utilize the results of a city's economic resilience status classification to determine the strategic direction of development.

## 2. Literature Review

### 2.1. Economic Resilience

The study of literature about economic resilience in this section strongly supports the author's opinion that the measurement of economic resilience must involve disturbance factors. Disturbance factors are then studied for their effects on the economic resilience of a system (city, province, country). The significant effects are studied by their impact on economic resilience status. If the economic resilience variable is in an unwanted condition after a specific degree of disturbance, the system is said to have no economic resilience, and vice versa. Below are some discussions on economic resilience. There are three ways of assessing the economic resilience of the regions. The first is based on the so-called "engineering" concept of resilience [12]. The supporters of this approach defined economic resilience as the ability of countries to withstand shocks and to recover quickly [13–15], and economic resilience as the ability of an area to respond and recover from disturbances [16]. Others define economic resilience as the ability of a region to recover successfully from

economic shocks [17]. The same thing is expressed as the ability of an area to recover from disturbances [18], and, within this approach, economic resilience refers to the ability of the local economic system to recover from an elastic shock [19]. Engineering resilience focuses upon forecasting the likelihood of catastrophic events and systemic breakdowns and their social and economic implications [20]. The paper from the World Bank states that macroeconomic resilience has two components: instantaneous resilience and dynamic resilience, which denote the ability to reconstruct and recover. The paper proposes rules of thumb to estimate macroeconomic and microeconomic resilience based on the relevant parameters in the economy [21]. The second approach to resilience is an "ecological" concept that emphasizes the magnitude of the disturbance that the region is able to absorb before it deviates from the original state [22]. The two approaches mentioned above are criticized because they do not sufficiently address the economic development perspective over the long term [23,24]. Resilience turns into the continuous ability to adjust to stress [25]. Opinions vary about the definition of resilience, and there is no mainstream approach to the measurement and expression of resilience. Therefore, there are no uniform strategies for strengthening the resilience of economies. This research adopts the second approach.

Economic resilience is the ability of individuals, communities, or countries to reduce vulnerability, to withstand shocks, and to recover quickly [26]. Economic resilience is the ability of cities to minimize potential losses due to disasters [27]. The economic resilience index uses microeconomic, mesoeconomic, and macroeconomic variables; the results of the study obtained a framework for obtaining an economic resilience index developed by [28]. Economic resilience is the ability of an area to anticipate, prepare to respond, and recover from disturbances [29]. Economic resilience is the ability of a region to recover successfully from economic shocks [30]. Regional resilience is the ability of an area to anticipate, prepare, respond to, and recover from disturbances [17].

Economic resilience is a systematic approach to reducing economic vulnerability and loss and improving critical disaster situations; the definition of regional economic resilience refers to the idea of the ability of the local economic system to recover from an elastic shock [18]. Research was undertaken under the title Theoretical and Empirical Analysis of Economic Resilience Index on economic resilience indexes applied to developing countries [31]. Regional economic resilience refers to the idea of the ability of the local economic system to recover from an elastic shock [9].

### 2.2. Chaos Theory

Chaos theory is a branch of mathematics focusing on the study of chaos states of dynamical systems whose apparently random states of disorder and irregularities are often governed by deterministic laws that are highly sensitive to initial conditions [32–34]. Chaotic behavior exists in many natural systems, including fluid flow, heartbeat irregularities, weather, and climate [35–37]. It also occurs spontaneously in some systems with artificial components, such as the stock market and economic conditions [38,39]. This research defines the word chaos according to common usage; "chaos" means "a state of disorder" [40,41]. The new chaotic system is shown to be multi-stable with simultaneous chaotic orientations [42].

### 2.3. Measurement Principle

The degree to which a test (such as a chemical, physical, or scholastic test) accurately measures what it is designed to measure is known as test validity. Validity is the benchmark criterion for assessing the quality of all measurement devices and procedures. The concept goes back at least to validity of measurements; if a measurement instrument is valid, it is measuring the right thing, what it is supposed to be measuring [43]. It also plays an important role in the ability of an item to discriminate between students who know the tested material and those who do not. The item will have low discrimination if it is so difficult that almost everyone gets it wrong or guesses, or so easy that almost everyone gets it right [44]. In psychometrics, there is the degree to which a test yields different

scores when applied to different criterion groups; specifically, the degree to which an item in an attitude scale, or the scale as a whole, yields different scores when it is applied to people holding different attitudes towards the attitude object in question. Discrimination power is a measure of the ability of a test to distinguish between two or more groups being assessed [45]. During test construction, items are often selected that have a discrimination index that is greater than a specified value [46]. According to [47], validity refers to how well a test measures what it is purported to measure. Item analysis as a set of measures and tests in order to assess the quality of the specific items of a testing instrument is discussed in [48]. One of the useful indicators is the discriminating power (DP).

The measurement of city economic resilience in this study also refers to the principles and spirit that underlie the measurement process as described above. The measurement method developed must be able to distinguish cities that have strong economic resilience and vice versa. This is related to the discriminating power of a measurement. In this study, discrimination power is a measure of the ability of a measurement to distinguish economic resilience among the 514 cities in Indonesia being assessed.

### 2.4. Individual Control Chart

In a general model for a control chart, let $w$ be a sample statistic that measures some quality characteristic of interest, and suppose that the mean of $w$ is $\mu_w$ and the standard deviation of $w$ is $\sigma_w$. Then the center line, the upper control limit (UCL), and the lower control limit (LCL) become:

$$
\begin{aligned}
UCL &= \mu_w + K\sigma_w \\
Center\ Line &= \mu_w \\
LCL &= \mu_w - K\sigma_w
\end{aligned}
\tag{1}
$$

$K$ is the "distance" of the control limits from the center line, expressed in standard deviation units [48]. This general theory of control charts was first proposed by Walter A. Shewhart, and control charts developed according to these principles are often called individual moving average control chart (I-Chart); they are used for this purpose [49]. Once the averages and limits are calculated, all of the individual data are plotted serially, in the order in which they were recorded. To this plot is added a line at the average value, x, and lines at the UCL and LCL values. Data accuracy is checked using the 30% acceptance sampling plan. A methodology that is designed specifically to perform the precedence analysis is developed. The methodology is modified I-Chart [50,51]. In this modified diagram, it is not fixed as in the ordinary individual control chart; it is necessary to calculate the moving average of the fixed average. The use of 3-sigma control limits gives good results in practice [48,50].

### 2.5. Economic Resilience Variable in Context of Disturbances Variable and Unwanted Conditions

In this case, the Z variable will distinguish cities' economic resilience in comparison to other cities. This is because the variable being compared is a buffer variable, which determines a city's resilience, rather than a concern variable. As a variable of economic resilience, Z must have a strong relationship with the disturbance variable and must also have a strong relationship with the concern variable. Without taking into account the level of disturbance, the economic resilience model is impossible. Thus, the economic resilience variable Z must exist together in the context of the disturber variable and Z as a control variable that functions as an absorber of the disturbance effect. Referring to the results of previous research on the economic resilience variable of cities in Indonesia, Z is vulnerable to increases in the price of Pertalite fuel variable and the US dollar currency [11].

A variable is declared feasible as a variable of economic resilience if it has a significant correlation with the disturbance variable in a certain relationship pattern. If a variable declared as a variable of economic resilience is not influenced by the disturbance variable, then the discourse on economic resilience becomes meaningless and irrelevant. Therefore, this paper examines the measurement model of city level economic resilience which takes

into account the effects of disturbances on the fall of the city's economic resilience status into the unwanted condition.

Economic resilience of a city requires the existence of unwanted conditions and a level of disturbance factors simultaneously; once again, the economic resilience model without taking into account the level of disturbance and unwanted conditions is an unrealistic model. Unwanted conditions are identified by determining the best relation model between Z and Pc variables. Z is the economic resilience variable (Z is the ratio between the original incomes of the region with the number of poor people in a city) and Pc is the concern variable (the ratio between gross regional domestic product and population). The unexplained variations are used as chaotic areas as the basis to state whether or not a city falls into an unwanted conditions area or not. Cities that fall into unwanted conditions are defined as cities that cannot bear receiving economic shocks [52].

Based on extensive literature research, the following is the formulation of the study's hypothesis:

Cities' economic resilience can be measured using a new simple mathematical model developed in Equation (2) that simultaneously accounts for both unwanted conditions and disturbance factors.

## 3. Materials and Methodology

### 3.1. Materials

This research used and analyzed time series data on an annual basis during year 2015 to 2019; all 514 Indonesian cities were studied. As a result, a census was conducted as part of this research. The following are the data that were utilized to apply a method for measuring a city's economic resilience.

(a) Considering variables of original local government revenue (PAD), there were 2570 observation units. PAD is revenue derived from regional income sources consisting of local taxes and others received.
(b) On the number of poor people, there are 2570 observation units.
(c) Population variables have 2570 observation units.
(d) Gross regional domestic product or PDRB in Indonesia.
(e) Considering variables of original local government revenue, there were 2570 observation units (PAD).
(f) Data of nine disturber variables: G1: price of Pertalite fuel oil, G2: premium fuel price, G3: gas price of 3 kg LPG, G4: gas price of 12 kg LPG, G5: basic electricity tariff of 900 VA subsidies, G6: basic electricity tariff 900 VA non-subsidized, G7: exchange rate of rupiah to US dollar, G8: Bank Indonesia reference interest rate, G9: consumer price index (CPI).

All data are collected from the Indonesian Central Bureau of Statistics' website (https://www.bps.go.id/) (accessed on 5 May 2021). The resilience variable analyzed is Z, the ratio between PAD and the number of poor persons, which is constructed using data (a) and (b). There were 2570 observation units for each variable G1, G2, . . . , G9, number of poor people, PDRB, and population. These variables' data were collected in order to construct a model of the relationships between economic resilience variables Z and disturbance variables G.

### 3.2. Methodology

A new methodology has been constructed to measure cities' economic resilience, taking into consideration both unwanted conditions and disturbance factors simultaneously. The economic resilience scores of the cities are expressed mathematically as follows:

**Theorem 1.** *The i-th Score:*

$$\frac{Z_u}{(1 - \Delta Z_i)} < Z_o < \frac{Z_u}{(1 - \Delta Z_{i+1})} \quad i = 1, 2, 3, 4, 5 \tag{2}$$

*where Zu is Z unwanted condition, $\Delta Z_i$ is the level of disturbance i and $\Delta Z_{i-1}$ is the level of disturbance i + 1, Zo is the initial condition of Z before being exposed to economic disturbances and Ze represents Z's final state when confronted with an economic disturbance. Z, the ratio of original local government revenue (PAD) to the number of poor people, is identified as economic resilience variable at the city level in Indonesia [16]. Given that Zu is the unwanted Z, a city's economic resilience level is measured by the value of Z, which indicates whether Z falls into unwanted conditions after being exposed to a certain level of disturbance. The proof of Equation (2) can be seen in the Equations (A1)–(A15) (Appendix A). While the proof of inequalities $\Delta Z_1$, $\Delta Z_2$, $\Delta Z_3$, $\Delta Z_4$, $\Delta Z_i$, $\Delta Z_5$ is conducted in four steps (Appendix B).*

### 3.2.1. Disturbance Level Design

The design of the intensity of the disturbance level and the criteria for unwanted conditions are critical in measuring economic resilience by taking disturbances into account. The most important thing in measuring economic resilience by taking into account disturbances is the design of disturbance levels. The intensity of the disturbance level and the criteria for unwanted conditions must be set properly in order to measure economic resilience by taking disturbances and unwanted conditions into consideration simultaneously. If the amount of disturbance is specifically set to be extreme, then all cities may be in a state where they are unable to tolerate economic shocks; measurement outputs are unable to differentiate a city's level of economic resilience. This is a failure in modeling the assessment of economic resilience because the measuring instrument fails to measure what it intended to measure; this is a validity issue that needs to be addressed. On the other extreme, if the level of economic disturbance is too mild, all cities are economically resilient. Again, we are unable to differentiate among city resilience.

As a result, defining the appropriate level of disturbance is a significant and strategic challenge in measuring economic resilience of cities.

The following is a construction of the measurement framework:

1.  Score 1 resilience 1: corresponds to the initial $Z_0$ value of cities that, after being exposed to level 1 disturbance, $\Delta Z_1$, have a final score of *Ze* that is higher than *Zu* as Z unwanted condition. This indicates that the resilience created will be measured in intervals form. It must be noted that the number of cities that can resist level 1 disturbance is infinite. The limitation is that if a city is disturbed by a level 2 disturbance, $\Delta Z_2$, cities will fall into an unwanted condition area, or mathematically $Ze < Zu$. The next resilience score is based on this reasoning.
2.  Score 2 resilience 2: refer to the initial $Z_0$ score of cities that when exposed to level 2 disturbance, $\Delta Z_2$, Ze's final score is above *Zu* but if shaken by level 3 disturbance, $\Delta Z_3$, cities will fall into unwanted condition areas, or mathematically Ze < Zu.
3.  Score 3 resilience 3: a city can only tolerate level 3 disturbances, $\Delta Z_3$, which are met with Ze > Zu conditions, and will fall into the unwanted condition, namely Ze < Zu if disturbed by level 4 disturbances.
4.  Score 4 resilience 4: a city can only tolerate level 4 disturbances, $\Delta Z_4$, which are met with Ze > Zu conditions, and will fall into the unwanted condition, namely Ze < Zu if disturbed by level 5 disturbances, $\Delta Z_5$.
5.  Score 5 resilience 5: a city can only tolerate level 5 disturbances, $\Delta Z_5$, which are met with Ze > Zu conditions, and will fall into the unwanted condition, namely Ze < Zu if disturbed by level 6 disturbances, $\Delta Z_6$.

The propositions are developed as a result of the measurement framework above. The empirical proof of the statements is shown in numerical results.

**Proposition 1.** *A measurement model will not be able to identify the level of economic resilience of a city if the level of disturbance is set to be too strong, because all cities may be in a state where they are unable to resist the economic disturbances received.*

**Proposition 2.** *If the level of economic disturbance is too mild, all cities will be found to be in a state of resilience and the measurement will fail to differentiate between cities' resilience.*

**Proposition 3.**

$$k < \frac{\left| \overline{\Delta Z} \right|}{\sigma_{\Delta Z}}$$

The disturbance level is designed as follows based on the experiment results: there are five levels of disturbance. Furthermore, the 1 to 5 disturbance level is designed using the statistical control chart method as shown in Equation (1) [51,52] to ensure that formula in Equation (A16) to formula in Equation (A20) is monotonically increasing. The constant k refers to the value obtained iteratively until the distribution of cities into five levels of economic resilience is reached. This is a sign of the success of measuring economic resilience in cities in Indonesia, as it enables cities to be differentiated from one another. Failure to select k shows a failure in determining the level of disruption, resulting in the inability to measure the city's economic resilience. The proof of proposition 3 is conducted from $\Delta Z_1 > 0$ and $\Delta Z_1 = \left| \overline{\Delta Z} \right| - k\sigma_{\Delta Z}$ (Appendix C).

**Theorem 2.** $\Delta Z_1$, $\Delta Z_2$, $\Delta Z_3$, $\Delta Z_4$, $\Delta Z_i$ $\Delta Z_5$ *Each respectively denote increasing disturbance level 1, disturbance level 2, disturbance level 3, disturbance level 4, and disturbance level 5 with the following disturbance level specifications: $\Delta Z_1$, $\Delta Z_2$, $\Delta Z_3$, $\Delta Z_4$, $\Delta Z_i$ $\Delta Z_5 < 1$. The proof of Theorem 2 can be seen in the Equations (A16)–(A20) (Appendix D).*

3.2.2. New Methodology for Measuring Economic Resilience

This study aims to measure a city's economic resilience level by simultaneously taking into account a disturbance model and unwanted conditions. In order to do that, the economic resilience of cities is measured by implementing a new, concise mathematical model that was developed in Equation (2). The equation is an integrated model that operationalizes the three preceding research results. This study integrates our previous studies [7,10,11] as a logical solution of the proposed methodology which combines a model of disturbance and unwanted conditions to measure economic resilience of a city. The following methods are derived as a logical consequence of the equation to obtain a city economic resilience score. It is provided here as a step-by-step explanation on how to conduct the implementation.

**Step 1.** Determine the economic resilience variable.

The most crucial step in measuring a city's economic resilience is establishing the economic resilience variable. A variable is considered appropriate to be used as an economic resilience variable if there is a significance correlation between it and the disturbance factors in a certain statistical model [10]. The discourse of economic resilience is regarded as meaningless and unnecessary if a variable designated as a variable of economic resilience is unchanged by the disturbance variable. The identification must be extensively searched since the required economic resilience variable that will differentiate a city's level of economic resilience applies nationally in Indonesia. In this stage, the best statistical model that explains the relationship between the disturber variable $\Delta G$ and disturbance variable $\Delta Z$ is selected. The best model between $\Delta G$ and Z must have a negative $\Delta G$ regression coefficient and a small $B_0$ intercept.

**Step 2.** Determine the best model that explains the relationship between the concern variable, Pc, with the economic resilience variable, Z, which was established in **step 1**. Pc stands for per capita income, which is computed as the ratio of gross regional domestic product to the population. It is necessary to find the best statistical model between Z and Pc that has a large Z regression coefficient, $B_1$, and a lower $B_0$ intercept. The output of this identification process is the percentage of variance unexplained which is required as input to the chaotic area identification procedure in the third steps.

**Step 3.** Determine unwanted conditions using chaotic area.

The percentage of variance unexplained from step 2 is used to identify the chaotic area and is considered to represent the unwanted conditions. Chaotic area [7] is obtained with $(U_Z, Pc(U_Z))$ coordinates or, in other words, unwanted conditions are limited by ordinate $Pc < Pc(U_Z)$ and abscissa $Z < U_Z$.

**Step 4.** Compute $|\overline{\Delta Z}|$ as absolute value of $\Delta Z$ average.

**Step 5.** Compute $\sigma_{\Delta Z}$ as standard deviation of $\Delta Z$.

**Step 6.** Substitute the results from steps 1 to 5 into Equation (2) with a specific k value determined through the iteration process. The methodology for measuring economic resilience at the city level is effective and completed if scores of economic resilience vary from one to five and are spread among all Indonesian cities. To put it another way, it is claimed that the proposed methodology is ineffective if only one score is recorded across all cities, because it cannot discriminate between the economic resilience of the city or is unable to measure what is intended to be measured. A workflow diagram is shown in Figure 1 to represent the steps in putting the methodology of measuring economic resilience into practice

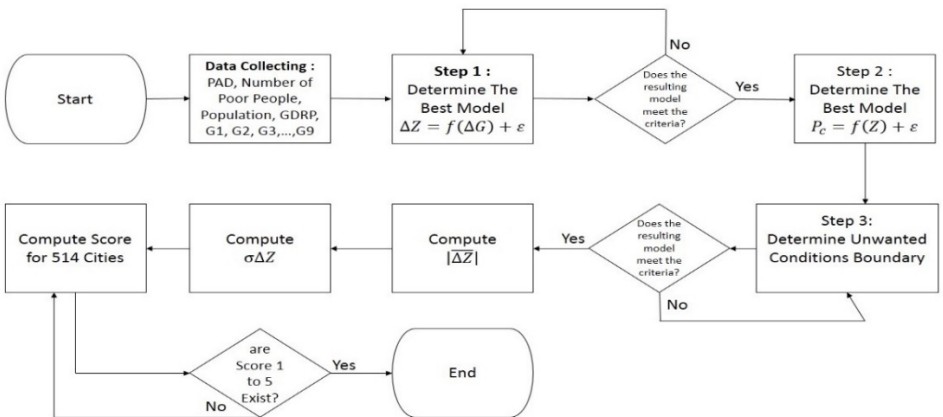

**Figure 1.** Workflow diagram describing sequence of steps involved in method for measuring a city's economic resilience.

## 4. Results and Discussion

### 4.1. Results of New Methodology for Measuring Economic Resilience

According to the previous section's explanation, the following results were produced using a new method for measuring a city's economic resilience. The iterative process was terminated after all 514 cities in Indonesia were classified into five economic resilience levels.

**Step 1.** The Economic Resilience Variable.

Our previous study [10] has identified that in Indonesia, due to the parsimonious analysis, the price of Pertalite gasoline (BBMPER) and the exchange rate for the US dollar (VA) perform as significant predictors and have a joint effect on the 73.63 percent fall in resilience variable Z. Here is the best statistical model between $\Delta G$ and $\Delta Z$ which has a negative $\Delta G$ regression coefficient and a small $B_0$ intercept.

$$\Delta Z = -0.3\ VA - 0.16\ BBMPER \tag{3}$$

According to the model of the relationship between Z and these two disturbance factors, the ratio of original regional income (PAD) to the number of poor people in a city is a city-level economic resilience variable, which applies nationally in Indonesia. Therefore, the research results show that it makes sense to use a single indicator for each city.

Why is poverty in isolated areas not taken into account? Since Indonesia has no area variability, this study does not address the effect. Isolated poverty will worsen the Gini index, simply because the money does not penetrate it. Gini does not determine resilience since there is not enough variation in Gini among Indonesian cities. This indicates that

the spatial distribution of poverty in the cities is constant. Therefore, we are unable to determine whether Gini is influential.

**Step 2.** Model of the relationship between economic resilience variables Z and concern variables Pc is expressed in Equation (4). The economic resilience variable Z is the ratio between PAD and the number of poor people. The data in points (b) and (e) are used to compute this ratio. The concern variable analyzed for each of the 514 cities is per capita income, Pc, which is the ratio of gross regional domestic product (PDRB) to population. The data in points (d) and (c) are used to compute this ratio. To analyze the model of the relationship between control variables Z and concern variables Pc, 2570 observation units for each variable PAD, number of poor people, PDRB, and population were collected. Pc averages IDR 35,559,642.42 each year, whereas Z averages IDR 7,919,249.12.

Advanced models related to non-linear estimation techniques are the best models for explaining the behavior of Pc as concern variables by resilience variable Z [11]. The method of piecewise linear regression analysis [49,51,53,54] is selected carefully. Utilizing the help of STATISTICA software [54], the results obtained are presented in Table 1 and Equation (4).

**Table 1.** Model of piecewise linear regression between Pc and Z (N = 2570).

|          | Const.B0   | Z     | Const.B0    | Z       | Breakpoint  |
|----------|------------|-------|-------------|---------|-------------|
| Estimate | 27,612,407 | 0.044 | 243,864,078 | −3.077  | 118,583,239 |

Note: Piecewise linear regression with breakpoint.

Dependent variable: Pc loss: least squares final loss: $230310895 \times 10^{10}$ R = 0.77966. Variance explained: 60.79%. The regression coefficients are obtained using the Rosenbrock pattern search estimation method [55]. Piecewise linear regression with breakpoint 118,583,239 can explain well the pattern of relationships between Z and Pc variables. The R-square of Equation (4) is 60.79 percent. It signifies that 40% of the variance (FVU) cannot be explained by changes in Z.

$$Pc = 27,612,407 + 0.04\ Z \tag{4}$$

**Step 3.** Unwanted condition boundary.

The fraction of variance unexplained (FVU) is defined as:

$FVU = 1 - R^2$

The FVU means that there is a 40% variation in Pc that cannot be explained by changes in Z. This statistic is not really good, but the correlation is too important to ignore; this 40% R-square is utilized as a chaotic area and is classified as an unwanted condition. In this area, high uncertainty occurs where city's authorities cannot predict changes in Pc based on changes in the Z as resilience variable.

The boundaries of the unwanted area are determined by looking at the chaotic boundary area with $(U_Z, Pc\ (U_Z))$ coordinates [10] or, in other words, the chaotic boundary is limited by ordinate $Pc < Pc\ (U_Z)$ and abscissa $Z < U_Z$. Chaotic area is located in the value of $U_Z$ less than IDR 5,097,592 and $Pc < Pc\ (U_Z) = 27,816,310.68$; thus, the coordinates of the chaotic boundary area are (5,097,592: 27,816,310.68).

The unwanted conditions are limited by ordinate $Pc < Pc\ (U_Z)$ and abscissa $Z < U_Z$. The unwanted condition is located in the value of UZ less than IDR 5,097,592 and $Pc < Pc\ (UZ) = 27,816,310.68$. Cities that are unable to resist economic shocks are classified as those that have fallen into unwanted conditions.

**Step 4.** Compute $\left|\overline{\Delta Z}\right| = 0.261196685$ as absolute value of $\Delta Z$.

**Step 5.** Compute $\sigma_{\Delta Z} = 0.240563055$ as standard deviation of $\Delta Z$.

**Step 6.** Substitute the results from step 1 to step 5 into Equation (2) with k = 1.08577.

In Indonesia, all 514 districts and cities are involved in the nationwide implementation of steps 1 through 6, which generated a score for economic resilience ranging from 1 to 5. Table 2 summarizes and displays the complete set of results.

**Table 2.** Frequency distribution of economic resilience scores for all Indonesian cities.

| Score | Z-Range | Frequency | Cities or Districts |
|---|---|---|---|
| 0 | Less than 5,104,680.91 | 69.18% | 1. District of Mesuji, 2. District of Lamongan 3. District of Kolaka, . . . , 1779. District of Tapin |
| 1 | 5,104,680.91–5,787,881.29 | 3.11% | 1. District of Tolitoli, 2. District of Dharmasraya, 3. District of Konawe Utara, . . . , 79. District of Kab ENDE |
| 2 | 5,787,881.29–6,899,796.63 | 4.24% | 1. District of Kotabaru, 2. District of Tanah Datar 3. District of Kayong Utara, . . . , 109. District of Deiya |
| 3 | 6,899,796.63–8,540,524.94 | 3.39% | 1. District of Murung Raya, 2. City of Jakarta Barat 3. District of Manokwari Selatan, . . . , 87. District of Bangka |
| 4 | 8,540,524.94–10,642,262.09 | 3.39% | 1. City of Bandar Lampung, 2. District of Belitung, 3. City of Bekasi1, . . . , 90. City of Bogor |
| 5 | More than 10,642,262.09 | 16.69% | 1. District of Belitung, 2. District of Melawi, 3. District of Simalungun, . . . , 425. District of Subang |

*4.2. Discussion*

The research hypothesis is well supported by the research findings that reveal the frequency distribution of economic resilience scores for all Indonesian cities. Cities' economic resilience can be measured using a new simple mathematical model developed in Equation (2) that simultaneously accounts for both unwanted conditions and disturbance factors. The frequency distributions presented in Table 2 provide proof that the proposed method has significant discriminating power. The ability of a measurement to distinguish between two or more groups being evaluated is known as discrimination power [53]. In this study, the term "discrimination power" refers to a measurement's ability to distinguish among the 514 Indonesian cities being assessed for their economic resilience. According to the theories and concepts discussed in the literature, the developed methodology must clearly differentiate among cities with high economic resilience and those without. This is correlated to the discrimination power of a measurement. The requirement has been met, as evidenced by the fact that the frequency distribution of economic resilience scores for all Indonesian cities is spread over all scores. The economic resilience measurement method is claimed to have a limited capacity for differentiation if the results of the scoring are either low, merely high, or bipolar. These events represent as a red flag that the disturbance level design is inaccurate. If the level of disturbance is deliberately designed to be very high, measurement outputs may not be able to distinguish between the levels of economic resilience of different cities, in which case all cities may be in a position where they are unable to withstand economic shocks. This is a failure in the modeling of the measurement of economic resilience since the measuring method does not measure what it was intended to measure. On the other hand, if the level of economic disturbance is designed to be very low, all cities are economically resilient. Once more, we cannot differentiate one city's resilience from another.

Since 69.18 percent of Indonesia's cities have confronted low-level disturbances in a non-resilient way, the results of Table 2 clearly show that Indonesia is one of the nations where low-level disturbances have a significant impact.

Because this research is so distinctive, comparison to other results and highlighting similarities and differences cannot be done. In particular, in Indonesia, there has never been any research that measures economic resilience at the city level. This study contrasts with previous research that measured economic resilience only based on an index, as done by previous researchers and the National Resilience Measurement Laboratory of the National Resilience Institute in Indonesia. Indexes are insufficient since they do not determine which disturbance factors have a significant effect on the city's economic resilience.

The method used to measure economic resilience in countries other than Indonesia differs significantly from the method used for this study. This study also contradicts existing

research that measures economic resilience primarily through disturbance models. To assess the significance of disturbances in degrading the city's economic resilience, a new concept is proposed. This concept relates to unwanted conditions that we have constructed. Only disturbances that cause a city to fall into an unwanted condition are considered disturbances. As a result, economic resilience suggests the existence of unwanted conditions and levels of disturbance. Two premises were developed in this research: first, the city's economic resilience without disturbance is meaningless, and second, the significance of disturbances is measured through their effect on the city's fall into unwanted conditions. Thus, a city is seen to be economically resilient if it does not fall into unwanted conditions when confronted with significant economic disturbances. Disturbance models and unwanted condition models are two critical factors that must coexist as determinants in assessing a city's economic resilience status.

Resilience is proven by the presence of the level of disturbance with an unwanted condition. The following analogy may help readers to comprehend this concept: a strong wind (level of disturbance) can destroy a building (unwanted condition). As a result, defining unwanted conditions is critical to the development of this economic resilience measurement. Since the issue of unwanted conditions in economic resilience research is unusual, this work develops a new algorithm that uses a chaotic area algorithm to find the unwanted condition.

## 5. Conclusions

Equation (2) represents the novelty of this study since it integrates a model of disturbance and unwanted conditions to measure a city's economic resilience. It is not reasonable to measure a city's economic resilience just on the basis of unwanted conditions or disturbance model. This study proposes methodology to determine a city's economic resilience status based on an assessment of the economic resilience variable, Z, as a substitute for the index method. If Z is in the unwanted condition when exposed to a specific level of disturbance, then the city will be considered economically non-resilient and the city's per capita income will decrease. These integrations were completed with the aid of the results from our earlier research on the identification of unwanted areas [7], which is positioned in step 3 of the methodology proposed, and the method for establishing Z as an economically resilient variable at the city level [10], which is arranged in step 1 of the methodology discussed in Section 3.

Six levels of economic resilience are identified using Equation (2) based on research of all 514 cities in Indonesia involving nine disturbance indicators and one economic resilience variable over a five-year observation period, 2015–2019. These six levels of economic resilience are identified using the five levels of disturbance generated using Equations (A16)–(A20). These are the six levels:

**Score 0 resistance 0:** This score belongs to a city that is not resistant to level 1 disturbances and will collapse into an unwanted condition if disturbed by a level 1 disturbance. In Indonesia, 69.18 percent of cities are classified as having low economic resilience (level 0).

**Score 1 resilience 1:** The city is only resilient to level 1 disturbances, and if it is disturbed by level 2 disturbances, it will fall into an unwanted condition. In Indonesia, as many as 3.11 percent of cities fall under the level 1 economic resilience category.

**Score 2 resilience 2:** The city is only resilient to level 2 disturbances, and if it is disturbed by level 3 disturbances, it will fall into an unwanted condition. In Indonesia, as many as 4.24 percent of cities fall into the level 2 economic resilience category.

**Score 3 resilience 3:** The city is only resilient to level 3 disturbances, and if it is disturbed by level 4 disturbances, it will fall into an unwanted condition. In Indonesia, as many as 3.39 percent of cities fall into the level 3 economic resilience class.

**Score 4 resilience 4:** The city is only resilient to level 4 disturbances and if it is disturbed by level 5 disturbances, it will fall into an unwanted condition. In Indonesia, as many as 3.39% of cities fall into the category level 4 economic resilience.

**Score 5 resilience 5:** The city is only resilient to level 5 disturbances, and if it is disturbed by level 6 disturbances, it will fall into an unwanted condition. In Indonesia, 16.69% of cities are classified as level 5 economically resilient.

### 5.1. Implications

What does a city get out of being "not resilient" according to the methodology? The city is not resilient if small disturbances make the city fall into an unwanted condition. What purpose does it serve? The purpose is to help the city government assess the security of the city so the government can take preventive actions to avoid the cities falling into unwanted condition. Referring to the results of this research, policymakers in Indonesia can concentrate on developing programs that aim to both increase regional original income and decrease poverty. The formula is necessary because it gives a way to assess a development program's effectiveness. The technique for chaotic areas that was established in this study is useful for evaluating economic resilience at the city/regency level. This method is developed to overcome the shortcomings of the index method commonly used internationally. Policymakers can utilize the results of the classification of a city's economic resilience status to determine the direction and strategy of growth based on the economic resilience score produced by the method in Equation (2). The development priority program for each city varies as a result of this research, depending on the degree of economic resilience of the city. Programs to enhance quality and supervision are available for cities with low economic resilience, and retention strategies are provided for cities with high economic resilience. How to deliver these results to those who can influence policy? Because of the solid linkages that exist between the government and universities (our university is the source of innovations for government), this is a remarkable experience in Indonesia.

### 5.2. Limitations

The suggested methodology may be performed globally, but the methodology's results only have local Indonesian relevance. We cannot conclude that the methodology is applicable universally because it has not been implemented internationally. Therefore, it is possible that our methodology would not find any indicators when applied in France. The Z is not really very important, though. The methodology which is used to derive Z is crucial. It is possible that Z only applies in Indonesia. Perhaps using the same methodology, Switzerland could discover P, Holland could discover Q, etc.

This study focuses on city economic resilience rather than provincial or even national economic resilience. Furthermore, this research must explore sensitivity analysis by performing experimentation to establish the maximum level of disturbance that will push a city's relative position toward a decline in economic resilience. Finding the threshold at which a city will collapse to unwanted conditions and be labeled as economically non-resilient is essential. The findings of the measurement can only be used to measure the city's level of economic resilience over the span of a year. The future predictability of the measurement results is unknown.

The paper's analysis of this study relates to processing secondary data. Therefore, the findings are unable to answer the query of how the loss of residuals can be observed in space. The study did not provide a solution to the field analysis question. The Z variable, or residual, used in this study is anything that cannot be observed in the field, just like inflation, the Gini coefficient, and per capita income, which are all calculated variables rather than observed variables.

**Author Contributions:** Conceptualization, T.P. and S.; methodology, T.P., S. and W.M.A.W.A.; data curation, T.P.; project administration, T.P. and S.; formal analysis, S. and W.M.A.W.A.; funding acquisition, S.; resources, S.; supervision, S.; validation, T.P., S. and Y.H.; visualization, T.P. and Y.H.; writing—original draft, T.P.; investigation, Y.H.; software, W.M.A.W.A.; writing—review and editing, W.M.A.W.A. All authors have read and agreed to the published version of the manuscript.

**Funding:** This APC was funded by Universitas Padjadjaran from Doctoral Dissertation Research Grant (RDDU), with a contract number 1595/UN6.3.1/PT.00/2021.

**Institutional Review Board Statement:** Not applicable.

**Informed Consent Statement:** Not applicable.

**Data Availability Statement:** Data is contained within the article.

**Acknowledgments:** The author would like to thank the Dean of the Faculty of Mathematics and Natural Sciences, Universitas Padjadjaran and the Directorate of Research and Community Service (DRPM), who have provided funding via the Universitas Padjadjaran Doctoral Dissertation Research Grant (RDDU), with a contract number: 1595/UN6.3.1/PT.00/2021.

**Conflicts of Interest:** The authors declare no conflict of interest.

## Appendix A

**Proofs of mathematical expressions in Equation (2).**

$$\Delta Z = \frac{Z_e - Z_o}{Z_o} \tag{A1}$$

$$\Delta Z . Z_o = Z_e - Z_o \tag{A2}$$

$$Z_e = Z_o + (\Delta Z . Z_o) \tag{A3}$$

$$Z_e = Z_o(1 + \Delta Z) \tag{A4}$$

When $Z_o$ is exposed to a certain amount of disturbance, it changes into $Z_i$ as expressed in Equation (A4). Equation (A4) is indeed a general model of change; because the disturbance is being investigated in this context, the relevant model of change is that $\Delta Z$ must be negative since it represents a specific level of disturbance or it is as if the problem is of price discounting. As a result of the consequence, Equation (A4) will now be rewritten as Equation (A5).

Note that $\Delta Z = 0$ is omitted because it does not reflect the nature of the disturbance.

It is important to note that there are two possible events in Equation (A4); the first event is:

$$Z_e > Z_u \tag{A5}$$

Equation (A5) demonstrates a city's ability to absorb varying degrees of economic shocks and disturbances, which is a formula that a city must achieve in order to be recognized as economically resilient when disturbed by an economic disturbance. The second event is:

$$Z_e < Z_u \tag{A6}$$

Equation (A8) states that a city is unable to resist the intensity of certain economic shocks or disturbances; this is the opposite of Equation (A5) and indicates that a city is not economically resilient.

Equation (A4) is substituted into Equation (A5), resulting in Equation (A7).

$$Z_o(1 - \Delta Z) > Z_u \tag{A7}$$

$$Z_o > \frac{Z_u}{(1 - \Delta Z)} \tag{A8}$$

In the same way, substituting Equation (A4) into Equation (A6) generates Equation (A9):

$$Z_o(1 - \Delta Z) < Z_u \tag{A9}$$

$$Z_o < \frac{Z_u}{(1 - \Delta Z)} \tag{A10}$$

The formula in Equation (2) is proven by using deductions from formulas in Equations (A8) and (A10).

If the increase in the intensity of the disturbance level $\Delta Z$ in Equation (2) is gradated into the disturbance levels of $\Delta Z_1$, $\Delta Z_2$, $\Delta Z_3$, $\Delta Z_4$, $\Delta Z_i$ $\Delta Z_5$ then the resulting economic resilience scores are as follows:

$$\frac{Z_U}{(1 - \Delta Z_1)} < Z_o < \frac{Z_U}{(1 - \Delta Z_2)} : \text{Score 1} \tag{A11}$$

$$\frac{Z_U}{(1 - \Delta Z_2)} < Z_o < \frac{Z_U}{(1 - \Delta Z_3)} : \text{Score 2} \tag{A12}$$

$$\frac{Z_U}{(1 - \Delta Z_3)} < Z_o < \frac{Z_U}{(1 - \Delta Z_4)} : \text{Score 3} \tag{A13}$$

$$\frac{Z_U}{(1 - \Delta Z_4)} < Z_o < \frac{Z_U}{(1 - \Delta Z_5)} : \text{Score 4} \tag{A14}$$

$$\frac{Z_U}{(1 - \Delta Z_5)} < Z_o < \frac{Z_U}{(1 - \Delta Z_6)} : \text{Score 5} \quad \square \tag{A15}$$

According to the formula at Equation (A11), the city is only resilient when it is disturbed to level 1 disturbances, and when it is disturbed by a level 2 disturbance, it will collapse into an unwanted condition. As shown in Equation (A11), cities with a score of 1 are resilient to level 1 disturbances but not to level 2 disturbances. A city with a score of 0 is not resilient to level 1 disturbances, meaning that it is economically vulnerable. The remaining scores are explained in the same way as the score 1 explanation. In Equation (A11), for instance, according to the formula in Equation (A15), the city is only resilient to level 5 disturbances, and if it is disturbed by a level 6 disturbance, it will break down into an unwanted condition.

**Appendix B**

**Proofs of Theorem 1** of inequalities $\Delta Z_1$, $\Delta Z_2$, $\Delta Z_3$, $\Delta Z_4$, $\Delta Z_i$, $\Delta Z_5$ is conducted in four steps:

**Step 1.** $\Delta Z_1$, $\Delta Z_2$

$$\Delta Z_2 > \Delta Z_1$$
$$\Delta Z_2 - \Delta Z_1 > 0$$
$$\left( \left| \overline{\Delta Z} \right| - \tfrac{1}{2} k \sigma_{\Delta Z} \right) - \left( \left| \overline{\Delta Z} \right| - k \sigma_{\Delta Z} \right) > 0$$
$$\left| \overline{\Delta Z} \right| - \tfrac{1}{2} k \sigma_{\Delta Z} - \left| \overline{\Delta Z} \right| + k \sigma_{\Delta Z} ) > 0$$
$$\tfrac{1}{2} k \sigma_{\Delta Z} > 0$$

Because k > 0 and $\sigma_{\Delta Z} > 0$ then $\Delta Z_1$, $\Delta Z_2$.

**Step 2.** $\Delta Z_2$, $\Delta Z_3$,

$$\Delta Z_3 > \Delta Z_2$$
$$\Delta Z_3 - \Delta Z_2 > 0$$
$$\left| \overline{\Delta Z} \right| - \left( \left| \overline{\Delta Z} \right| - \tfrac{1}{2} k \sigma_{\Delta Z} \right) > 0$$
$$\left| \overline{\Delta Z} \right| - \left| \overline{\Delta Z} \right| + \tfrac{1}{2} k \sigma_{\Delta Z} > 0$$
$$\tfrac{1}{2} k \sigma_{\Delta Z} > 0$$

Because k > 0 and $\sigma_{\Delta Z} > 0$, then it is proven that $\Delta Z_2$, $\Delta Z_3$,

**Step 3.** $\Delta Z_3$, $\Delta Z_4$

$$\Delta Z_4 > \Delta Z_3$$
$$\Delta Z_4 - \Delta Z_3 > 0$$
$$\left| \overline{\Delta Z} \right| + \tfrac{1}{2} k \sigma_{\Delta Z} - \left| \overline{\Delta Z} \right| > 0$$
$$\tfrac{1}{2} k \sigma_{\Delta Z} > 0$$

Because k > 0 and $\sigma_{\Delta Z} > 0$, then it is proven that $\Delta Z_3$, $\Delta Z_4$,

**Step 4.** $\Delta Z_4$, $\Delta Z_5$

$$\Delta Z_5 > \Delta Z_4$$
$$\Delta Z_5 - \Delta Z_4 > 0$$
$$(|\overline{\Delta Z}| + k\sigma_{\Delta Z}) - (|\overline{\Delta Z}| + \tfrac{1}{2}k\sigma_{\Delta Z}) > 0$$
$$|\overline{\Delta Z}| + k\sigma_{\Delta Z} - |\overline{\Delta Z}| - \tfrac{1}{2}k\sigma_{\Delta Z} > 0$$
$$\tfrac{1}{2}k\sigma_{\Delta Z} > 0$$

Because k > 0 and $\sigma_{\Delta Z} > 0$, then $\Delta Z_4$, $\Delta Z_5$. □

**Appendix C**

**Proofs of Proposition 3.**

$$\Delta Z_1 > 0$$
$$\Delta Z_1 = |\overline{\Delta Z}| - k\sigma_{\Delta Z}$$
$$|\overline{\Delta Z}| - k\sigma_{\Delta Z} > 0$$
$$|\overline{\Delta Z}| > k\sigma_{\Delta Z}$$
$$k\sigma_{\Delta Z} < |\overline{\Delta Z}|$$
$$k < \frac{|\overline{\Delta Z}|}{\sigma_{\Delta Z}} \quad □$$

**Appendix D**

**Proofs of Theorem 2.** The 1st level of disturbance is represented by the lowest control limit as defined in Equation (A16).

$$\Delta Z_1 = |\Delta \overline{Z}| - k\sigma_{\Delta z} \tag{A16}$$

$$\Delta Z_2 = |\Delta \overline{Z}| - \frac{1}{2}k\sigma_{\Delta z} \tag{A17}$$

The disturbance level two in Equation (A17) is one level higher than the lowest control limit in Equation (A16).

$$\Delta Z_3 = |\Delta \overline{Z}| \tag{A18}$$

Third level of disturbance is expressed by central limit as formulated in Equation (A18).

$$\Delta Z_4 = |\Delta \overline{Z}| + \frac{1}{2}k\sigma_{\Delta z} \tag{A19}$$

The fourth level of disruption is designed one level higher than the central limit as expressed in Equation (A19).

$$\Delta Z_5 = |\Delta \overline{Z}| + k\sigma_{\Delta z} \tag{A20}$$

The highest control limit, as described in Equation (A20), acts as the fifth level of disturbance.

$|\Delta \overline{Z}|$: absolute mean value of $\Delta Z$, where $|\Delta \overline{Z}| > 0$.

$\sigma_{\Delta Z}$: standard deviation of $\Delta Z$, $\sigma_{\Delta Z} > 0$.

$K$: a constant value derived by iteration in order to differentiate among various levels of disturbance. $k > 0$. The iteration procedure is used to obtain an acceptable value of k as a logical consequence of propositions 1 and 2. Another implication of proposition 1 and 2 is the generation of proposition 3 and 4. □

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
