# Peer review of "Developing New Method in Measuring City Economic Resilience by Imposing Disturbances Factors and Unwanted Condition"

_computation, doi:10.3390/computation10080135_

Round 1

Reviewer 1 Report

The research problem undertaken by the authors is interesting and rarely addressed in the literature. The large empirical material collected during the research deserves praise.

However, in my opinion, there are 4 main aspects to improve:

1.      The literature review is very rich and structured. However, from the literature review, it is not very clear what are the research hypotheses.

2.      There is very limited section of the discussion of results. The obtained results should be referenced to other results, compare to indicate similarities and differences.

3.      In the conclusive paragraph, the description of the implications of the research and its main limitations should be included.

4.      Technical mistakes: line 37, 46, 52-53, 654

Author Response

Dear reviewers,

We are quite appreciative of the insightful evaluation that was provided. The advice you provided helped us enormously. Your feedback motivates us to make improvements. Although we are conscious that we might not be able to meet all of your expectations, we do our best to take all of your recommendations into consideration. We are thrilled to have the chance to learn from you; this is a priceless opportunity. The comments you leave inspire us to work: “The research problem undertaken by the authors is interesting and rarely addressed in the literature”, “The large empirical material collected during the research deserves praise”, and “The topic is very interesting”. Any and all feedback provides encouragement to keep working on getting better.

Much appreciation. Greetings and best wishes to everyone.

Cordially yours

Reviewer

Comment

Corrective actions

Reviewer 1

1.     The literature review is very rich and structured. However, from the literature review, it is not very clear what are the research hypotheses

the research hypotheses has been formulated in line.221-223

2.     There is very limited section of the discussion of results. The obtained results should be referenced to other results, compare to indicate similarities and differences.

The discussion of results has been improved significantly

 Because this research is so unique, it is difficult to establish the findings to those of other studies or point out their parallels and differences. There has never been any study that evaluates economic resilience at the city level, considered simultaneously disturbance factor and chaotic area particularly in Indonesia.

3.     In the conclusive paragraph, the description of the implications of the research and its main limitations should be included

The research's implications are discussed, and its primary limitations are highlighted and listed.line 551 -572

4. technical mistakes: line 37, 46, 52-53, 654

technical mistakes Line 37, 46, 52-53, 654 has been corrected

Reviewer 2 Report

First of all, the topic is very interesting 

However, the paper would benefit from some minor addition, such as;

The intro section is a bit lengthy and would be beneficial to reduce this section.

The references would be required to cover the at least the last five years.

The methodology section is not explained what are you planning to conduct. A separate method would have a great addition to the reader to identify the information needed. 

The analysis and conclusion are both very well written 

Author Response

Dear reviewers,

We are quite appreciative of the insightful evaluation that was provided. The advice you provided helped us enormously. Your feedback motivates us to make improvements. Although we are conscious that we might not be able to meet all of your expectations, we do our best to take all of your recommendations into consideration. We are thrilled to have the chance to learn from you; this is a priceless opportunity. The comments you leave inspire us to work: “The research problem undertaken by the authors is interesting and rarely addressed in the literature”, “The large empirical material collected during the research deserves praise”, and “The topic is very interesting”. Any and all feedback provides encouragement to keep working on getting better.

Much appreciation. Greetings and best wishes to everyone.

Cordially yours

Reviewer

Comment

Corrective actions

Reviewer 2

1.      The intro section is a bit lengthy and would be beneficial to reduce this section.

The introduction section has been shortened and reworked.

2.     The references would be required to cover the at least the last five years

Even though we completely updated the references to be from the previous five years or more, some of them are still relevant to the topic, so we still include it.

3.      The methodology section is not explained what are you planning to conduct. A separate method would have a great addition to the reader to identify the information needed. 

 The methodology part is included and explained in more detail, and it includes a workflow diagram outlining the procedures required in a method for calculating a city's economic resilience. It is provided here a step-by-step explanation on how to conduct the implementation

Line 249

4.      The analysis and conclusion are both very well written

The remark is extremely motivating, thank you.

Reviewer 3 Report

Excerpts from the introduction are from an earlier paper. 

Please add a research workflow diagram. Numbering and names of individual blocks of the diagram should be in accordance with the description in the text. 

The literature review contains mathematical formulas rewritten from other works I suggest not to include them in the article. Instead, please indicate the bibliographic source properly. 

Please describe in more detail the conclusions that follow from the authors' previous work that influenced the current research work. In what sense is the current work a simple continuation of previous work?

The conclusions should be formulated in a more applied and practical way:

(a) what does a city get out of being "not-resilient" according to this methodology? What purpose does it serve?

(b) how to present these results to people with policy influence? (In my opinion, it is inaccessible to prove under the conditions of an isolated example, so the dwod cannot be considered universal outside the boundary conditions)

(c) how can the loss of residuals be observed in space, and does a single indicator per city make sense if the density of poor people is not included (what about isolated "neighborhoods of poverty")?

(d) how does a city with a residual of 1 differ from one with a residual of 3 qualitatively?

All proofs of mathematical theorems should be placed in the Appendix

English is colloquial in places + inconsistency in the notation of numbers/digits - once in words, once in Arabic numerals

Author Response

Dear reviewers,

We are quite appreciative of the insightful evaluation that was provided. The advice you provided helped us enormously. Your feedback motivates us to make improvements. Although we are conscious that we might not be able to meet all of your expectations, we do our best to take all of your recommendations into consideration. We are thrilled to have the chance to learn from you; this is a priceless opportunity. The comments you leave inspire us to work: “The research problem undertaken by the authors is interesting and rarely addressed in the literature”, “The large empirical material collected during the research deserves praise”, and “The topic is very interesting”. Any and all feedback provides encouragement to keep working on getting better.

Much appreciation. Greetings and best wishes to everyone.

Cordially yours

Reviewer

Comment: Excerpts from the introduction are from an earlier paper.

We have cited pertinent information from a previous paper.

Reviewer 3

1.      Please add a research workflow diagram. Numbering and names of individual blocks of the diagram should be in accordance with the description in the text. 

Workflow diagram describing sequence of steps involved in method for measuring a city's economic resilience are presented in Figure 1.line 369-372

2.      The literature review contains mathematical formulas rewritten from other works I suggest not to include them in the article. Instead, please indicate the bibliographic source properly. 

There are no longer any mathematical formulas in the literature review that have been copied and pasted from other works. Instead, we have properly cited the source in the bibliography.

3.      Please describe in more detail the conclusions that follow from the authors' previous work that influenced the current research work. In what sense is the current work a simple continuation of previous work?

These integrations were completed with the aid of the results from our earlier research on the identification of unwanted areas [7], which is positioned in step 3 of the methodology proposed, and the method for establishing Z as an economically resilient variable at the city level [10], which is arranged in step 1 of the methodology .Line 521-523

4.      The conclusions should be formulated in a more applied and practical way:

(a) what does a city get out of being “not-resilient” according to this methodology? What purpose does it serve?

(b) how to present these results to people with policy influence? (In my opinion, it is inaccessible to prove under the conditions of an isolated example, so the dwod cannot be considered universal outside the boundary conditions)

© how can the loss of residuals be observed in space, and does a single indicator per city make sense if the density of poor people is not included (what about isolated “neighborhoods of poverty”)?

(d) how does a city with a residual of 1 differ from one with a residual of 3 qualitatively?

All proofs of mathematical theorems should be placed in the Appendix

English is colloquial in places + inconsistency in the notation of numbers/digits – once in words, once in Arabic numerals

The research made on the particular case of Indonesia can be generalized using the fraction of variance unexplained (FVU) statistics as a measure a chaotic area.  As long as the FVU can be determined from any statistical model, it is always possible to set uncertainty areas which is unpredictable areas or chaotic areas as unwanted conditions for policy makers in various systems. The method in this research was developed to over-come the weaknesses of the method of measuring economic resilience using the index approach.

(a)   are explained in line 552-555

(b)   are explained in line 568-571

(c)   are explained in line 391-400 and 589-594

(d)   are explained in line 589-594

All proofs of mathematical theorems have placed in the Appendix
